# Glaucoma-Associated CDR1 Peptide Promotes RGC Survival in Retinal Explants through Molecular Interaction with Acidic Leucine Rich Nuclear Phosphoprotein 32A (ANP32A)

**DOI:** 10.3390/biom13071161

**Published:** 2023-07-22

**Authors:** Carsten Schmelter, Kristian Nzogang Fomo, Alina Brueck, Natarajan Perumal, Sascha D. Markowitsch, Gokul Govind, Thomas Speck, Norbert Pfeiffer, Franz H. Grus

**Affiliations:** 1Department of Experimental and Translational Ophthalmology, University Medical Center, Johannes Gutenberg University, 55131 Mainz, Germany; cschmelter@eye-research.org (C.S.); kristianfomo@yahoo.de (K.N.F.); alina.brueck@web.de (A.B.); nperumal@eye-research.org (N.P.); norbert.pfeiffer@unimedizin-mainz.de (N.P.); 2Department of Urology and Pediatric Urology, University Medical Center, Johannes Gutenberg University, 55131 Mainz, Germany; sascha.markowitsch@unimedizin-mainz.de; 3Institute of Physics, Johannes Gutenberg University, 55131 Mainz, Germany; gogovind@uni-mainz.de (G.G.);

**Keywords:** glaucoma, neuroprotection, ANP32A, proteomics, mass spectrometry

## Abstract

Glaucoma is a complex, multifactorial optic neuropathy mainly characterized by the progressive loss of retinal ganglion cells (RGCs) and their axons, resulting in a decline of visual function. The pathogenic molecular mechanism of glaucoma is still not well understood, and therapeutic strategies specifically addressing the neurodegenerative component of this ocular disease are urgently needed. Novel immunotherapeutics might overcome this problem by targeting specific molecular structures in the retina and providing direct neuroprotection via different modes of action. Within the scope of this research, the present study showed for the first time beneficial effects of the synthetic CDR1 peptide SCTGTSSDVGGYNYVSWYQ on the viability of RGCs ex vivo in a concentration-dependent manner compared to untreated control explants (CTRL, 50 µg/mL: *p* < 0.05 and 100 µg/mL: *p* < 0.001). Thereby, this specific peptide was identified first as a potential biomarker candidate in the serum of glaucoma patients and was significantly lower expressed in systemic IgG molecules compared to healthy control subjects. Furthermore, MS-based co-immunoprecipitation experiments confirmed the specific interaction of synthetic CDR1 with retinal acidic leucine-rich nuclear phosphoprotein 32A (ANP32A; *p* < 0.001 and log2 fold change > 3), which is a highly expressed protein in neurological tissues with multifactorial biological functions. In silico binding prediction analysis revealed the N-terminal leucine-rich repeat (LRR) domain of ANP32A as a significant binding site for synthetic CDR1, which was previously reported as an important docking site for protein-protein interactions (PPI). In accordance with these findings, quantitative proteomic analysis of the retinae ± CDR1 treatment resulted in the identification of 25 protein markers, which were significantly differentially distributed between both experimental groups (CTRL and CDR1, *p* < 0.05). Particularly, acetyl-CoA biosynthesis I-related enzymes (e.g., DLAT and PDHA1), as well as cytoskeleton-regulating proteins (e.g., MSN), were highly expressed by synthetic CDR1 treatment in the retina; on the contrary, direct ANP32A-interacting proteins (e.g., NME1 and PPP2R4), as well as neurodegenerative-related markers (e.g., CEND1), were identified with significant lower abundancy in the CDR1-treated retinae compared to CTRL. Furthermore, retinal protein phosphorylation and histone acetylation were also affected by synthetic CDR1, which are both partially controlled by ANP32A. In conclusion, the synthetic CDR1 peptide provides a great translational potential for the treatment of glaucoma in the future by eliciting its neuroprotective mechanism via specific interaction with ANP32A’s N terminal LRR domain.

## 1. Introduction

Glaucoma is described as a group of optic neuropathies, which is mainly caused by progressive dysfunction and degeneration of retinal ganglion cells (RGCs) accompanied by gradual loss of vision. Advanced age, genetics as well as an elevated intraocular pressure (IOP) are major risk factors for developing glaucoma, which represents 80 million affected patients worldwide as the most common cause of blindness [1,2]. To date, IOP-managing strategies are still the gold standard in glaucoma therapy, accomplished by the administration of topical eye drops or by the implementation of operational procedures [3]. Nevertheless, many patients are refractory to IOP-lowering medications, and most surgeries only achieve a stabilization of the IOP in the short-term [4]. Accordingly, the currently available therapies delay disease progression in spite of low adjusted IOPs and do not directly interfere with the neurodegenerative component of glaucoma. Furthermore, up to 30% of glaucoma patients exhibit an IOP in the physiological range (IOP = 10–21 mmHg) [5], termed normal-tension glaucoma, illustrating the versatile and multifactorial characteristics of this chronic eye disease. All these findings demonstrate the urgent need for new strategic directions in glaucoma research and the development of new innovative medical therapeutics which provide direct RGC neuroprotection instead of solely managing clinical symptoms.

Novel immunotherapeutic strategies provide excellent requirements to target specific molecular structures which interfere with the pathological mechanism of the respective disease [6,7]. Thereby, immunotherapeutics possess a wide variety of modes of action ranging from receptor blocking or inhibition of signaling transmission to much more regulatory functions such as immunomodulation or cell cycle coordination. Particularly, antibodies, as well as antibody-derived peptides (encoding the hypervariable complementarity-determining region, CDR), offer promising therapeutic approaches for the treatment of glaucoma in vivo and in vitro by inducing direct RGC neuroprotection [8,9,10,11]. Recent findings of our group confirmed, for instance, that monoclonal anti-HMGB1 enhanced the viability of RGCs in vivo by modulating the retinal RNA metabolism as well as the cytokine secretion [9]. We suppose that these neuroprotective effects were caused by interfering with the HMGB1-mediated inflammatory responses and by targeted modulation of its pathogenic biological function. In accordance with that, an antibody-derived immunopeptide of our group triggered RGC neuroprotection ex vivo by antagonizing mitochondrial dysfunction and through activation of the cellular antioxidant defense system [8]. In this case, the molecular mode of action was based on targeted inhibition of the mitochondrial serine protease HTRA2, which represents a key player in maintaining mitochondrial homeostasis and neuronal cell survival [8,12].

The present study focuses on the acidic leucine-rich nuclear phosphoprotein 32A (ANP32), which belongs to a family of highly conserved proteins and regulates as a multifunctional protein many biological processes, including apoptosis, cell cycle progression (tumor suppressor), neurogenesis, transcriptional regulation as well as protein phosphorylation [13]. ANP32A is highly expressed in neurological tissues such as the cerebellum or the cerebral cortex and is located in the cytoplasm as well as in the nucleus [14]. It has also been reported to function as an extracellular mediator suppressing apoptosis of hepatocellular carcinoma cells [15] and was also associated with the oncogenesis of many other cancer types [16,17,18]. Regarding the molecular structure, ANP32A consists of an N-terminal leucine-rich repeat (LRR) domain and a C-terminal low complexity acidic region (LCAR) containing up to 70% aspartic (D) and glutamic (E) acid residues [19,20]. Particularly, the LRR domain shows a curved shape favoring protein-protein interactions (PPI), whereas the LCAR seems to be an important feature for chromatin binding and microtubule regulations [19,20,21]. Due to its diverse biological activities, dysfunction of ANP32A was already associated with other neurodegenerative-related diseases such as Alzheimer’s disease [22] or Spinocerebellar ataxia [23] and was also discussed as a potential therapeutic target [24,25,26]. However, the functional role of ANP32A is largely unexplored in glaucoma so far and might serve as attractive molecular target structure for RGC neuroprotection in glaucoma research.

The CDR1 peptide SCTGTSSDVGGYNYVSWYQ (homologous to *IGLV2-11*03*) was identified first as a potential biomarker candidate in primary open-angle glaucoma (POAG) patients [27] and offers promising properties as a therapeutic agent. Due to that reason, the main objective of the present study was to evaluate the neuroprotective potential of the synthetic CDR1 peptide on RGCs ex vivo and to characterize its molecular function via interaction with ANP32A. To address these questions, we performed peptide-based immunoprecipitation experiments combined with quantitative proteomic analyses (mass spectrometry and Western blot analyses). Furthermore, we employed computational docking analysis to elucidate the molecular binding mechanism from synthetic CDR1 to ANP32A. The results of this project provide important information about the medical application of ANP32A as a potential drug target in glaucoma therapy as well as for the treatment of other age-related neurodegenerative diseases.

## 2. Materials and Methods

### 2.1. Synthetic CDR Peptides

The CDR1 peptide (SCTGTSSDVGGYNYVSWYQ) was identified first as a potential biomarker candidate in the serum of POAG patients and was significantly lower expressed in systemic IgG molecules compared to healthy control subjects [27]. The original peptide comprised in total >30 amino acids (SVSGSPGQSVTISCTGTSSDVGGYNYVSWYQQHPGK). Due to problems during peptide synthesis by the manufacturer as well as insufficient peptide purity, we decided to restrict the peptide length to <20 aa. The remaining sequence encoded the CDR1 sequence motif (underlined part) flanked by 5 aa to each side representing the adjacent framework regions (FR1 and FR2). Due to experiences from previous studies [8,11], we supposed that particularly the CDR1 sequence motif determines the biological function and activity of the peptide. The company Synpeptide Co. Ltd. (Shanghai, China) chemically synthesized the peptides with a purity of >90%. The peptide synthesis was performed as follows: SCTGTSSDVGGYNYVSWYQ (without modification) and Biotin-[Acp]-SCTGTSSDVGGYNYVSWYQ. The unmodified synthetic peptides were evaluated as potential glaucoma medications in the retina organ culture model (see method Section 2.2). The peptides with the N-terminal Biotin-[Acp] modification were immobilized on magnetic streptavidin beads for immunoprecipitation experiments (see method Section 2.3).

### 2.2. Retina Organ Culture Model and Immunohistochemical Staining

The retina organ culture model is frequently used in our laboratory for RGC neurodegeneration, which is elicited by axotomy of the optic nerve and characterized by progressive RGC loss during 24 h of cultivation [8,10]. It mimics fundamental characteristics of glaucoma and is routinely used to evaluate the neuroprotective effects of potential glaucoma medications [8,10]. The retina organ culture was prepared from freshly removed eye bulbs of the house swine (*Sus scrofa domestica* Linnaeus, 1758), which has been extensively described in former studies [8,10]. The removed eye bulbs were transported within 2 h on ice from the slaughterhouse to the laboratory facilities and were further prepared under sterile conditions. For this experiment, 5 × 5 mm retina-retinal pigment epithelium (RPE) tissue compounds were assembled and cut in the dorsal periphery above the visual streak to ensure homogeneous distribution of RGCs in each experiment. Thereby, the retina and RPE were merged into an in vivo orientation, with the photoreceptor layer lying face down and the RGC layer oriented upwards. Neurobasal A medium supplemented with 2% B27, 1% N2, 0.8 mM L-alanyl-L-glutamine, and 1% penicillin/streptomycin was used for the cultivation of the retinal explants for 24 h at 37 °C and 5% CO_2_. The tissue compounds were either untreated as control explants (CTRL) or treated with 50 µg/mL and 100 µg/mL of synthetic CDR1 (SCTGTSSDVGGYNYVSWYQ), respectively. Both concentrations were revealed from a previous experiment (see Appendix A) and selected as starting point for the present study. A concentration of 25 µg/mL of synthetic CDR1 did not show any beneficial effects compared to the untreated CTRL, whereas a concentration of 150 µL/mL indicated a potential saturation effect (see Appendix A). The next day, the control and peptide-treated retinae (*n* = 6 for each group) were carefully removed from the RPE and subsequently fixed in 4% paraformaldehyde (PFA) for 30 min at RT. The immunohistochemical staining for the detection of RGCs was performed by a standard operation protocol (SOP) as previously described [8]. For this staining, we used the goat anti-Brn3a (ratio: 1:250; Santa Cruz Biotechnology, Dallas, TX, USA) as the primary antibody, followed by labeling with the fluorescent secondary dye antibody Alexa Fluor 568 donkey anti-goat (ratio: 1:400; Thermo Fisher Scientific, Rockford, IL, USA, Cat. No.: A-11057). In addition, each retinal flat-mount was stained with 1:2500 4′,6-Diamidin-2-phenylindol (DAPI; Thermo Fisher Scientific, Rockford, IL, USA, Cat. No.: D1306) in PBS. The microscopic analyses were performed with the fluorescent microscope Axio Observer Z1 (Carl Zeiss AG, Oberkochen, Germany) using a 20-fold magnification objective lens. Eleven high-resolution fluorescent images were taken from each retinal flat-mount at different positions providing in a total of 66 fluorescent images for each group (CTRL and CDR1: 50 and 100 µg/mL). The fluorescent images were randomized, and the number of RGCs (Brn3a^+^ cells) was manually counted by experienced laboratory workers using the software package ImageJ [23]. Brightness and contrast were adjusted in each image to facilitate the counting of the Brn3a^+^ cells and to extrapolate the numbers to RGC/mm^2^ for each group.

### 2.3. Peptide-Based Immunoprecipitation

The peptide-based immunoprecipitation experiment was performed to identify protein interaction partners of synthetic CDR1 in the retina. Therefore, we used retinal protein homogenate from the house swine (*Sus scrofa domestica* Linnaeus, 1758) for the enrichment, which was prepared as described elsewhere in detail [8]. The prepared protein homogenate was stored in 5 mg aliquots at −20 °C before further processing. For peptide immobilization, we employed Pierce^TM^ Streptavidin Magnetic Beads (Thermo Fisher Scientific, Rockford, IL, USA, Cat. No.: 88816) for the immunoprecipitation experiments. Thus, 50 µL of magnetic beads solution was washed twice with 200 µL of phosphate-buffered saline (PBS) using a magnetic stand and subsequently labeled with 80 µg of Biotin-[Acp]-SCTGTSSDVGGYNYVSWYQ for 1 h at RT. Simultaneously, magnetic control (CTRL) beads were labeled with 0.5 mg/mL biotin solution (Thermo Fisher Scientific, Rockford, IL, USA, Cat. No.: 29129), which will be used as reference values for the normalization of the quantitative proteomic data. After chemical peptide immobilization, the bead fractions (*n* = *3* per group) were washed twice with 200 µL of PBS, followed by the addition of 5 mg retinal protein homogenate to each assay (5 mg in 200 µL PBS, protein concentration: 25 µg/µL). The mixtures were incubated at 4 °C overnight with gentle mixing. The next day, the unbound protein fractions were discarded, and both bead groups ± CDR1 were washed three times with 300 µL of PBS. The remaining attached proteins were eluted with 100 µL of Pierce^TM^ IgG Elution buffer (pH 2.0; Thermo Fisher Scientific, Rockford, IL, USA, Cat. No.: 21028) from the magnetic beads ± CDR1 and subsequently neutralized with 10 µL of 1 M Tris HCl solution (pH 8.5). The protein eluates of each group (CTRL and CDR1) were evaporated in the speed vacuum concentrator (SpeedVac; Eppendorf, Darmstadt, Germany) until dryness at 30 °C and stored at −20 °C before further processing (see method Section 2.5).

### 2.4. Molecular Dynamics (MD) Simulations and Virtual Docking Analysis

The initial configurations of the CDR1 peptide sequence were modeled using PyMOL [28], and all MD simulations were performed using GROMACS 2021 [29]. The peptide was equilibrated for 2 ns in both NVT and NPT ensembles using SPC/E water as the explicit solvent [30]. MD simulations were performed for 200 ns, and to obtain better sampling data, five independent 200 ns simulations starting from random velocities were carried out for the CDR1 peptide. The clusters were rendered using VMD 1.9.4 [31], and all the figures were plotted using Python version 3.11. Autodock Vina 1.1.2 was used to predict potential CDR1-specific binding regions of ANP32A (PDB protein structure: 4XOS) [32]. The peptide secondary structure used for the docking binding energy calculations was received from the MD simulations.

### 2.5. In-Solution Trypsin Digestion

The SOP for the mass spectrometric (MS) analysis of peptide-based immunoprecipitation experiments (see method Section 2.3) is described elsewhere in detail [8]. The sample preparation protocol for the MS analysis of the retina organ culture ± CDR1 treatment was slightly modified. At first, the retina organ culture model was performed as described in method Section 2.2, and the retinal tissues subsequently snap-frozen in liquid nitrogen (*n* = 3 for each group; CTRL and CDR1 100 µg/mL). Afterward, the frozen retinae ± CDR1 treatment was subjected to further homogenization and tryptic digestion as described in a previous publication [8]. Therefore, the retinal proteins were extracted in T-PER™ buffer (Thermo Fisher Scientific, Rockford, IL, USA, Cat. No.: 78510) using the Precellys^®^ 24 homogenizer (VWR, International GmbH, Darmstadt, Germany) and subsequently rebuffered in 200 µL of PBS using an Amicon 3 kDa centrifugal filter device (Millipore, Billerica, MA, USA, Cat. No.: UFC500396). After the determination of the protein concentration of each sample, 10 µg of protein was evaporated in the SpeedVac until dryness at 30 °C. Each protein sample was resolved in 30 µL of 10 mM ammonium bicarbonate (ABC) with 3 µL of 20 mM dithiothreitol (DTT) in 10 mM ABC. All samples were incubated at 56 °C for 30 min to reduce the disulfide bonds of the proteins. As the next step, 3 µL of 40 mM iodoacetamide (IAA) in 10 mM ABC was added for the alkylation of the released cysteine residues. Hence, all samples were incubated for 30 min at RT in the dark, followed by enzymatic digestion with 10 µL of trypsin solution (1 mg/mL in 10 mM ABC 10% acetonitrile ((ACN); Promega; Madison, WI, USA, Cat. No.: V5111) at 37 °C overnight. The next day, the digestion was quenched with 10 µL of 0.1% formic acid (FA), and all samples evaporated in the SpeedVac until dryness at 30 °C. Prior to MS analysis, peptide purification was carried out with SOLAµ^TM^ HRP SPE spin plates (Thermo Fisher Scientific, Rockford, IL, USA, Cat. No.: 60209-001) [8] and stored as dried as well as concentrated samples at −20 °C before further experimental procedure.

### 2.6. Mass Spectrometric (MS) Analysis

The MS measurements were performed with a Hybrid Linear Ion Trap-Orbitrap MS system (LTQ Orbitrap XL; Thermo Fisher Scientific, Rockford, IL, USA), which is a well-established technology for quantitative proteomics in our laboratory [8,27]. In the case of the peptide-based immunoprecipitation experiments (see method Section 2.3), the MS system was online coupled to a capillary HPLC system as described elsewhere [27]. Respectively, experimental settings for this HPLC-MS analysis are listed in previous studies [8]. For the other quantitative proteomic analysis (see method Section 2.2 and Section 2.5), the LTQ Orbitrap XL MS operated with the EASY-nLC 1200 system (Thermo Fisher Scientific, Rockford, IL, USA), which was combined with the PepMap C18 column system (75 mm × 500 mm; Thermo Fisher Scientific, Rockford, IL, USA). Solvent A for the nanoLC instrument comprised 0.1% FA in water, and solvent B consisted of 0.1% FA in 80% ACN. In total 2 µL of each sample (peptide concentration: 0.125 µg/µL) were injected into the nanoLC system and were measured twice to receive technical replicates. The chromatographic separation of the peptides was performed within 200 min using a flow rate of 0.3 µL/min with the following solvent gradient: 5–30% B (0–160 min), 30–100% B (160–180 min) and 100% B (180–200 min). The MS data acquisition was performed with a resolution of 30,000 in the positive ion mode, and the target automatic gain control (AGC) was set to 1 × 10^6^ ions. The internal lock mass calibration was set to 445,120,025 *m*/*z* (poly-dimethyl cyclosiloxane). Dynamic exclusion (DE) of the MS system was enabled during data acquisition with the following settings: Repeat count = 1, repeat duration = 30 s, exclusion list size = 100, exclusion duration = 300 s, and exclusion mass width of ± 20 ppm. After each FTMS scan, the five most intense precursor ions were fragmented in the ion trap using a normalized collision energy of 35%. All MS proteomics data of this study have been deposited to the ProteomeXchange Consortium via the PRIDE [33] partner repository with the dataset identifier PXD038373.

### 2.7. Quantitative Analysis of the Proteomic MS Data

The acquired tandem MS data were analyzed with bioinformatics software MaxQuant v. 1.6.17.0 (Max Planck Institute for Biochemistry, Martinsried, Germany). The reviewed SwissProt databases with the taxonomies *Homo sapiens* (date: 13 April 2021, number of sequences: 20,408) and *Sus scrofa* (date: 13 April 2021, number of sequences: 1439) were used for protein identifications and quantification. Due to the low public availability of proteomic data for the house swine (*Sus scrofa*) [34], we included the species-related human proteomic database (*Homo sapiens*) for this quantitative analysis. The following software settings were used for the database search: peptide ion mass tolerance of ± 30 ppm, ion fragment mass tolerance of 0.5 Da, tryptic cleavage, maximum of two missed cleavage sites, carbamidomethylation as fixed modification, acetylation (N-terminal protein) and oxidation as variable modifications. In addition, the “match between run” function was enabled, and all protein identifications were filtered with a false discovery rate (FDR) < 1%.

### 2.8. Statistics and Bioinformatics

The output data of the MaxQuant analysis were statistically analyzed with the software program Perseus v. 1.6.15 (Max Planck Institute for Biochemistry, Martinsried, Germany). As the first step, proteins identified as contaminants, reversed hits, or “only identified by site” were removed from the data table. Afterward, the intensity values of the filtered proteins were log2-transformed. Reliable protein identifications with ≥2 unique peptides had to be present in at least three biological replicates of one study group (CTRL or CDR1-treated group) and were employed for data imputation and further statistical analysis. Imputation of missing protein abundances was based on the normal distribution of the data for each replicate (width: 0.3, down-shift: 1.8). For the statistical analysis of the peptide-based immunoprecipitation experiment (see method Section 2.3), only missing values of the control bead group (CTRL) were imputed in accordance with previous publications [8,35]. After data imputation, two-sided *t*-test statistics for pairwise comparison were used to identify significantly changed proteins between the experimental groups (*p* < 0.05). For data visualization, the protein intensities were standardized by Z-score followed by illustration in the Heatmap using Euclidean for hierarchical clustering. Further statistical analyses (*t*-test statistics or ANOVA for multiple group comparison) as well as a graphical presentation of the data were performed with Statistica version 13 (StatSoft; Tulsa, OK, USA). Furthermore, significantly changed protein markers were screened for molecular-biological annotations as well as signaling functions using biological database STRING version 11.5 (Search Tool for the Retrieval of Interacting Genes/Proteins) and Ingenuity Pathway Analysis software v. 1-04 (IPA, Ingenuity QIAGEN; Redwood City, CA, USA) [8].

### 2.9. Western Blot Analysis

To evaluate the protein phosphorylation and histone acetylation events in the retinae ± CDR1 treatment, we performed a Western Blot analysis. Therefore, 50 µg of retinal lysate + CDR1 treatment were loaded onto 10-well NuPAGE 12% Bis-Tris mini gels (Thermo Fisher Scientific, Rockford, IL, USA, Cat. No.: NP0341BOX) under reducing conditions, which were incorporated in the XCell SureLock™Mini-Cell Electrophoresis System (Invitrogen, Carlsbad, CA, USA). In addition, 10 µL of the SeabluePlus 2 Pre-Stained Protein Standard (Thermo Fisher Scientific, Rockford, IL, USA, Cat. No.: LC5925) was used as molecular weight reference. For 1D gel electrophoresis, the NuPAGE^TM^ MES SDS Running Buffer 20X (Thermo Fisher Scientific, Rockford, IL, USA, Cat. No.: NP0002) was used according to the supplier’s protocol and separated at 150 V for 1.5 h at 4 °C. Then, the protein was transferred to PVDC membranes for 1 h at 100 V. After blocking with 10% non-fat dry milk for 1 h; the membranes were incubated overnight with the following primary antibodies: Phospho-(Ser/Thr) Phe antibody (source: rabbit, dilution: 1:1000; Cell Signaling, Frankfurt am Main, Germany, Cat. No.: 9631), PAN (histone) acetylation antibody (source: mouse, dilution: 1:1000; Proteintech GmbH, Planegg-Martinsried, Germany, Cat. No.: 66289-1-Ig) and GAPDH antibody (source: mouse, dilution: 1:000, Cell Signaling, Frankfurt am Main, Germany; Cat. No.: 97166). As secondary antibodies, we used HRP-conjugated horse-anti-mouse IgG or goat-anti-rabbit IgG (dilution 1:10,000; Cell Signaling, Frankfurt am Main, Germany, Cat. No.: 7076 and 7074). Afterward, the SignalFire™ ECL Reagent (Cell Signaling, Frankfurt am Main, Germany, Cat. No.: 6883) was used to visualize the proteins according to the supplier’s protocol and detected with a Sapphire Imager (Azure Biosystems, Munich, Germany). The AlphaView SA software package v. 3.4 (ProteinSimple, San Jose, CA, USA) was used for the densitometric analysis. The protein band intensities were normalized to the expression profiles of GAPDH and calculated as percentage distribution. For multiple antibody incubations, the PVDC membrane was treated with Western blot recycling stripping buffer (10X; Alpha Diagnostic International, San Antonio, TX, USA, Cat. No.: 90101) according to the supplier’s protocol.

## 3. Results

### 3.1. Effects of Synthetic CDR1 on RGCs Ex Vivo

At first, we were interested in the potential neuroprotective effects of the synthetic CDR1 peptide (SCTGTSSDVGGYNYVSWYQ) on the viability of RGCs ex vivo. For the evaluation of this assumption, we used the retina organ culture of the house swine (*Sus scrofa domestica*), which is a well-established model system, to investigate RGC neurodegeneration [8,10]. The neurodegeneration in this model was elicited by axotomy of the optic nerve from the retina resulting in reproducible and significant loss of RGCs during 24 h of cultivation ex vivo. It provided excellent requirements to mimic glaucomatous-like conditions ex vivo and was frequently applied to deliver first hints for neuroprotection of potential glaucoma medications, particularly immunotherapeutics [8,10,11]. Accordingly, the retinal explants were either treated with 50 µg/mL and 100 µg/mL of synthetic CDR1 peptide or remained untreated, serving as the control group (CTRL) (see Figure 1 and Appendix A). After cultivation, the retinal explants were prepared for immunohistochemical Brn3a staining, which is a reliable molecular marker for the fluorescent microscopic identification of RGCs [36]. Figure 1A shows exemplary high-resolution images of the retinal explants ± treatment indicating an increasing RGC density (number of identified Brn3a^+^-RCGs is indicated in each image by *n*) with ascending CDR1 peptide concentration (from 50 to 100 µg/mL). The quantitative analysis of the microscopic data (see Figure 1B) revealed that retinal explants treated with 50 µg/mL of synthetic CDR1 (274 ± 30 RGCs/mm^2^) showed significantly higher viability of RGCs compared to untreated CTRL explants (352 ± 62 RGCs/mm^2^, *: *p* < 0.05). This neuroprotective effect could be clearly enhanced by ascending the CDR1 peptide concentration to 100 µg/mL resulting in a 31% significantly higher RGC viability (399 ± 52 RGCs/mm^2^) compared to the untreated control group (274 ± 30 RGCs/mm^2^, ***: *p* < 0.001).

### 3.2. Peptide-Based Immunoprecipitation of Retinal Protein Interaction Partners

To elucidate the neuroprotective mode of action of the synthetic CDR1 peptide (SCTGTSSDVGGYNYVSWYQ), we screened for potential interaction partners in the retina proteome using CDR1-specific immunoprecipitation experiments. This state-of-the-art MS-based technology enabled the identification of direct protein interaction partners of largely unexplored biological compounds and was already successfully applied by our group for target structure (biomarker) identification [8,11]. For this reason, we immobilized the synthetic CDR1 peptide on magnetic beads by biotin-streptavidin conjugation and performed a protein enrichment from retinal protein homogenate. For differentiation and exclusion of unspecific protein binders, we also considered a biotin-labeled control (CTRL) bead group in this experiment. After enrichment of the protein binders, the eluate fractions of both bead groups (CTRL and CDR1) were subjected to further proteomic MS analysis. Employing the statistical analysis of the quantitative proteomic data, we verified the significantly enhanced enrichment of acidic leucine-rich nuclear phosphoprotein 32A (ANP32A) by the CDR1-labeled magnetic beads in comparison to unlabeled CTRL bead group (*n* = 3, *p* < 0.001 and log2 fold change > 3, see Figure 2 and Appendix A). This result indicates that the synthetic CDR1 peptide seems to interact physically with retinal protein ANP32A.

### 3.3. MD Simulations and Binding Prediction Analyses of Synthetic CDR1

For better clarification of the molecular binding mechanism for this specific peptide–protein interaction, we achieved the first MD simulations in silico to reveal important secondary structure information about synthetic CDR1 (SCTGTSSDVGGYNYVSWYQ) (see Figure 3 and Appendix A). The most predominant peptide structure from the equilibrium MD simulations (see Figure 3A,B) possessed a *β*-hairpin-like structure in the center position of the peptide partially formed by the CDR1 sequence motif S(6)-SDVG-G(11). Furthermore, the center hairpin formation is connected to a turn on either side, followed by a random coil at the N-terminus and a short helical-like structure at the C-terminus. Particularly, due to the *β*-hairpin-like structure, the peptide exhibits high flexibility in the central region (see Figure 3C) arranged with two flexible portions at both edges (N- and C-terminal). In consideration of the specific secondary structural features of synthetic CDR1, we accomplished further peptide binding prediction analyses in silico to ANP32A (PDB protein structure: 4XOS), which was identified as a potential target protein by previous experiments. As a main result, the virtual docking analysis predicted a significant binding site of CDR1 to the N-terminal region of ANP32A (see Figure 3D,E), which is determined as the leucine-rich repeat (LRR) domain of ANP32A [20]. Especially, the highly flexible *β*-hairpin motif at the center portion of the peptide seems to favor the molecular interaction with the acidic LRR domain of ANP32A. Accordingly, these findings of the in silico calculations are in line with the experimental observations and support a potential molecular interaction between synthetic CDR1 and retinal target protein ANP32A.

### 3.4. Quantitative Proteomic Analysis of CDR1-Treated Retinal Explants

For elucidation of the molecular modes of action of the synthetic CDR1 peptide (SCTGTSSDVGGYNYVSWYQ), we employed a quantitative MS-based proteomic analysis of the retinal explants ± CDR1 treatment (*n* = 3 for each group). Due to increased neuroprotection on RGCs ex vivo in a concentration-dependent manner of synthetic CDR1, we selected a peptide concentration of 100 µg/mL for this experiment. In summary, 1040 proteins were identified in the retina of both experimental groups (CTRL and CDR1) using a false discovery rate (FDR) <1% (see Appendix A). Up to 3% of these proteins showed a significant level change between both groups (*p* < 0.05), indicating 12 high abundant and 13 low abundant protein markers in the CDR1-treated retinal explants (100 µg/mL) compared to the untreated CTRL group (see Figure 4A and Table 1 and Table 2). With respect to these results, we also identified CDR1-specific target protein ANP32A in the proteomic analysis indicating slightly diminished expression levels in the peptide-treated group compared to CTRL (see Figure 4B and Appendix A), even if this effect was not supported by statistical significance (*p* = 0.105). However, particularly pyruvate metabolism-related proteins (e.g., pyruvate carboxylase, PC and pyruvate dehydrogenase E1 α, PDHA1), as well as cytoskeleton/membrane-associated proteins (e.g., moesin, MSN and endophilin-A1, SH3GL2), were found with significantly higher abundances in CDR1-treated explants compared to CTRL. Otherwise, proteomic subunits of the retromer complex (e.g., vacuolar protein sorting-associated protein 26B, VPS26B and vacuolar protein sorting-associated protein 29, VPS29), as well as direct ANP32A-interacting proteins (e.g., nucleoside diphosphate kinase A, NME1 and serine/threonine-protein phosphatase 2A activator, PPP2R4), were significantly lesser expressed by CDR1 treatment compared to untreated CTRL (see Figure 5 and Table 2). Furthermore, IPA functional annotation analysis revealed Acetyl-CoA biosynthesis I, mitochondrial dysfunction, granzyme A signaling, oxidative phosphorylation, as well as mechanisms of viral exit from host cells as the top five most CDR1-induced signaling pathways (see Table 3).

### 3.5. Western Blot Analysis

Due to the important regulatory function of ANP32A in phosphoprotein signaling [37,38] and gene transcription [13], we investigated if there are any qualitative as well as quantitative changes regarding protein phosphorylation and histone acetylation events in retinal tissues ± CDR1 treatment (100 µg/mL). Therefore, we performed a Western blot analysis of the respective retinal protein lysates ± CDR1 and screened for protein phosphorylation sites at serine (Ser) or threonine (Thr) residues and for acetylation motifs at lysine (Lys) residues on histones (see Figure 6). The densitometric analysis of the data revealed that CDR1-treated retinal tissues tended to show a slight increase in protein phosphorylation events compared to untreated CTRL explants but was not supported by statistical significance (*p* = 0.18, see Figure 6B). Also, histone acetylation was clearly increased by CDR1 treatment compared to untreated CTRL, as indicated by the enhanced chemiluminescent signal in the CDR1-treated replicates (around 14 kDa; see Figure 6A). However, a convincing quantitative analysis of this marker was not feasible due to increased chemiluminescent background emission of the membrane.

## 4. Discussion

Immunotherapeutic strategies have revolutionized the treatment of many diseases ranging from neurodegenerations to various cancer types [39,40]. Thereby, the term ‘immunotherapeutics’ encompasses the medical application for a wide range of immune-related drug compounds, which are used for immunomodulation, immunization, check-point inhibition, receptor blocking, and other mechanism-based therapies. Especially, peptide-based immunotherapeutics are of great interest for pharmaceutical drug development due to the almost unlimited possibilities to improve their physicochemical properties (e.g., hydrophobicity) in order to guarantee the best pharmacokinetics characteristics (e.g., plasma half-lives or absorption) for various medical applications [41]. To date, there is no appropriate treatment option that directly interferes with the neurodegenerative molecular mechanism of glaucoma, and specific drug target molecules for therapy are still missing. Antibody-derived immunopeptides might represent an attractive therapeutic strategy to satisfy this urgent medical need and to provide new innovative perspectives in glaucoma therapy.

The present study results proved that the synthetic CDR1 sequence motif SCTGTSSDVGGYNYVSWYQ induced neuroprotective effects on RGCs ex vivo in a concentration-dependent manner (see Figure 1). Thereby, a concentration of 50 µg/mL of the synthetic CDR1 peptide significantly increased the RGC viability by about 22% compared to the untreated CTRL group (*p* < 0.05), whereas an augmented concentration of 100 µg/mL significantly enhanced the RGC neuroprotection of about 31% in comparison to CTRL (*p* < 0.001). Furthermore, the synthetic CDR1 peptide showed a high affinity for the acidic leucine-rich nuclear phosphoprotein 32A (ANP32A) in the retinal proteome (see Figure 2) and might elicit its neuroprotective activities via this specific protein interaction. Nevertheless, we required higher amounts of the synthetic CDR1 peptide (50–100 µg/mL) to induce RGC neuroprotection in the present study compared to the previous one [8], which might reflect the protein abundances of the respective target molecule in the retina. In the previous study, we used the CDR1 sequence motif: ASGYTFTNYGLSWVR (25 µg/mL) to induce RGC neuroprotection, which targets the mitochondrial serine protease HTRA2 [8] and inhibits its proteolytic activity [12]. ANP32A is a highly expressed protein in many neurological tissues (e.g., cerebral cortex) [14] and was also identified with high abundances in the retina in several MS-based studies of our group [11,42]. Target protein HTRA2, in contrast, was never identified in these MS analyses (without enrichment), indicating much lower expression levels in the retina, which are below the limit of detection of the MS. To support this, the human protein atlas provides a relative expression of ANP32A with 107 nTPM (normalized transcripts per million) in the retina and a relative abundance of HTRA2 with only 2.5 nTPM in the same tissue [43] indicating a fold change of 43:1 between both target molecules. This might explain the required CDR1 peptide concentration of ≥50 µg/mL (SCTGTSSDVGGYNYVSWYQ) for RGC neuroprotection ex vivo to provide sufficient drug target saturation and appropriate binding efficiency to ANP32A. Moreover, many peptides were reported to penetrate the cell membrane by direct translocation or by receptor-mediated endocytosis (e.g., clathrin-mediated endocytosis, which might also be a potential mechanism of synthetic CDR1 to enter the retinal cells [44].

The CDR1-specific target protein ANP32A is, in many aspects, a multifunctional protein participating in apoptosis, cell cycle regulation, transcriptional events as well as protein phosphorylation [13]. Interestingly, the selective knockdown or down-regulation of ANP32A ameliorated the cognitive deficits as well as the synaptic plasticity in different experimental animal models for Alzheimer’s disease (AD) and was discussed as a potential molecular marker for neuroprotection [24,25,26]. These beneficial effects were caused by the suppression of the formation of the inhibitor of the histone acetyltransferase (INHAT) complex, in which ANP32A represents an essential key component. The INHAT complex inhibits histone acetylation by a mechanism called histone-masking, in which the access of acetyltransferases is blocked by steric hindrance to the histones. Particularly, the C-terminal low complexity acidic region (LCAR) of ANP32A favors the direct interaction with histones and is absolutely required to suppress histone acetylation by the INHAT complex [13,45]. Accordingly, the ANP32A-induced hypoacetylation of histones as important epigenetic regulation seems to be responsible for the cognitive decline in AD in vivo [24,25]. In the present study, we observed a clear increase in histone acetylation events by synthetic CDR1 compared to untreated CTRL explants (see Figure 6A), which might explain the beneficial effects on RGCs ex vivo. On the other hand, synthetic histone-binding peptides (derived from the LCAR of ANP32A, region: 151–180 aa) were successfully used to treat acute myeloid leukemia by inhibiting histone acetylation of multiple target genes followed by cell cycle arrest and increased apoptosis of the leukemia cells [46]. However, the virtual docking analyses predicted that the synthetic CDR1 peptide specifically interacts with the N-terminal leucine-rich repeat (LRR) domain of ANP32A (see Figure 3). This observation leads to the assumption that the LRR domain of ANP32A seems to be also an important binding site for gene transcription regulation and might promote the RGC neuroprotection mechanism ex vivo.

The quantitative MS analysis of the retinal explants ± CDR1 treatment (100 µg/mL) revealed that 12 proteins were significantly up-regulated and 13 proteomic markers were significantly down-regulated between both experimental groups (*p* < 0.05, see Figure 4A and Table 1 and Table 2). In accordance, the protein levels of target protein ANP32A were slightly diminished in the CDR1-treated group compared to untreated CTRL, even not statistically significant (*p* = 0.105, see Figure 4B). Nevertheless, direct protein interaction partners nucleoside diphosphate kinase A (NME1) and serine/threonine-protein phosphatase 2A activator (PPP2R4) were also significantly decreased in the CDR1-treated retinal explants compared to CTRL, showing a similar trend like ANP32A (see Figure 5). Especially, the N-terminal LRR domain of ANP32A represents an important binding site for PPIs [19,20], which is possibly blocked by the synthetic CDR1 peptide in this case. However, the major function of ANP32A-interacting protein NME1 is to control the intracellular nucleotide homeostasis and is also involved in various other cellular processes such as metastasis suppression and DNA damage [47]. Due to its multifunctional activities, NME1 is possessed many regulatory domains, such as the nucleoside-diphosphate kinase (NDPK), the protein-histidine kinase, and the 3′-5′ exonuclease [47,48]. Interestingly, NME1 was already associated with the invasion and metastasis of various cancer types [49,50] but also induced neurite growth and neuroprotection in different model systems for Parkinson’s disease (PD) in vivo and in vitro [51,52]. In terms of ANP32A, it is also already known that both molecules are important subunits of the ER-associated oxidative stress response (SET) complex, which plays a critical role in the Granzyme A (GzmA)- or ROS-mediated nuclear DNA damage response [53,54] (see Table 3). Thereby, the SET complex is arranged by nucleases (e.g., NME1), chromatin-modifying proteins (e.g., ANP32A), and a DNA binding protein unit (e.g., HMGB2) that recognizes distorted DNA. The authors Radic’ et al. (2020) [55] demonstrated that NME1 translocates from the cytoplasm to the nucleus after radiation-induced DNA damage in vitro and was supposed to promote DNA damage repair. However, the connection between the SET complex and neurodegenerations is still unexplored but might represent an important effector mechanism of the CDR1-induced RGC neuroprotection. Possibly, it might also influence histone acetylation and gene transcription regulation as observed in our present study (see Figure 6A). Nevertheless, this finding should be confirmed in future studies since the Western blot showed a high chemiluminescent background emission and just delivers a hint for increased histone acetylation events by synthetic CDR1. Also, the metabolic key regulator CoA is also known to inhibit the NDPK activity of NME1 via covalent and non-covalent interactions [56], whose biosynthesis itself was the most affected signaling pathway in the CDR1-treated retinal explants (see Table 3).

The other ANP32A-interacting protein PPP2R4 represents the regulatory subunit of the protein phosphatase 2A (PP2A), which catalyzes the selective removal of phosphate groups from serine (S) and threonine (T) residues [57,58]. Thus, PPP2R4 determines the correct folding of the catalytic domain of PP2A and defines its substrate specificity. Generally, protein phosphorylations play a pivotal role in cell signaling, protein activity regulation, gene transcription as well as metabolism and have been associated with many neurodegenerative diseases so far, such as AD and PD [57,58]. In this context, target protein ANP32A was already described as an active inhibitor of PP2A (termed as I_1_^PP2A^) about twenty years ago [37]. Remarkably, the inhibition of PP2A was exclusively elicited by the LRR domain of ANP32A [38], which was identified as the preferred binding site of synthetic CDR1 in the present study. However, Latarya et al. (2012) [59] reported that the activities of PP2A as well as other phosphatases (PP2C and PTP), were significantly increased in the aqueous humor of glaucoma patients compared to non-glaucomatous cataract patients. However, hyperphosphorylated protein markers (e.g., Tau) were already associated with the pathogenesis of glaucoma [60]. Moreover, *PP2A* knockout mice showed impaired synapse transmission and synapse plasticity [61], highlighting the essential role of proper protein phosphorylation regulation for neuronal cell survival and homeostasis. Conclusively, the CDR1-induced down-regulation of PPP2R4 (subunit of PP2A) might be beneficial for RGC viability during glaucomatous-like conditions ex vivo. In accordance, our results support the increased occurrence of protein phosphorylation events by synthetic CDR1 (see Figure 6A,B), which might reflect the reduced enzymatic activity of PP2A.

Another low abundant marker protein in the CDR1-treated group was cell cycle exit and neuronal differentiation protein 1 (CEND1), which is highly expressed along the neuronal lineage ranging from neuronal stem or precursor cells to mature neurons [62]. CEND1 has an important function during neurogenesis by regulating cell cycle progression/exit and neuronal differentiation [62,63]. Interestingly, it has been observed that there is a significant down-regulation of CEND1 in the retina after optic nerve injury [64] or after blast exposure-mediated ocular damage [65] in vivo, respectively. The authors Siddiqui et al. (2014) [64] assumed that low CEND1 expression might indicate abnormal RGC functioning by promoting apoptosis and mitochondrial dysfunction. Also, *CEND1^−/−^*-deficient mice possess impaired cognitive functions and cellular damage in the brain clarifying its essential neuroprotective role [66]. In contrast, our present study confirmed a CDR1-induced down-regulation of CEND1, which could be accordingly considered unfavorable for RGC viability. Nevertheless, we speculate that this observation might indicate lesser neuroinflammatory responses in the CDR1-treated retinae, possibly caused by reduced cellular stress signaling (e.g., by protein phosphorylation regulation). In accordance with that, the expression levels of retromer complex subunits (e.g., vacuolar protein sorting-associated protein 26B and 29, VPS26B and VPS29) were also significantly decreased in the retina by CDR1 treatment and regulated other important key mechanisms for neuroprotection via the endolysosomal recycling pathway [67,68].

On the other hand, we also identified protein markers that were significantly highly expressed in the CDR1-treated retinae compared to CTRL (see Figure 4 and Table 1), amongst other enzymes of the acetyl-CoA biosynthesis from pyruvate (e.g., pyruvate dehydrogenase E1 α, PDHA1) and the pyruvate metabolism (pyruvate carboxylase, PC). PDHA1 is a subunit of the pyruvate dehydrogenase complex (PDH) and catalyzes the reaction from pyruvate to acetyl-CoA and CO_2_ [69]. Subsequently, acetyl-CoA can be further metabolized into ATP in the mitochondria or into other organic compounds and provides an important cellular source of energy. With respect to glaucoma, it is already known that an elevated IOP leads to a significant decline of pyruvate in the retina in vivo, accompanied by a disturbed glucose metabolism [70]. In accordance, pyruvate was reported to inhibit neuroinflammation and function as a ROS scavenger [71]. Interestingly, Sato et al. (2020) [72] showed that the inhibition of the pyruvate dehydrogenase kinase (PDK) improved the viability and the energetic function of RGCs in vivo. Thereby, the enzymatic activity of PDH is regulated by PDK, which inhibits its biological function by phosphorylation. Hence, the increased expression of PDHA1 (subunit of PDH) seems to have a great significance on the CDR1-induced RGC neuroprotection ex vivo and also highlights the relation between the pathogenesis of glaucoma and phosphoprotein cell signaling. Furthermore, the enzyme PC catalyzes the reaction from pyruvate to oxaloacetate, which serves as an essential intermediate for various biochemical pathways such as gluconeogenesis, the citric acid cycle, and glutamate/glutamine synthesis [73]. Particularly, astrocytes are the major source for the synthesis of glutamine in neurological tissues, which is needed to maintain neurotransmitter homeostasis [74]. Impaired astrocyte metabolism was already observed in different AD animal models resulting in decreased metabolic fluxes by PC and reduced glutamine synthesis [75,76]. Also, in glaucoma, it is well known that an elevated IOP leads to disturbed retinal glutamate/glutamine cycle activities and is closely connected to the apoptosis of RGCs [77,78]. However, it seems to be that the increased metabolic processing of pyruvate ameliorates the viability of RGCs ex vivo and might be elicited through the specific CDR1-ANP32A interaction. In addition, we also identified the cytoskeleton-regulating protein moesin (MSN) with high abundance in CDR1-treated retinae compared to CTRL, whose phosphorylation state was also associated with dysfunction of the blood-retina barrier in an experimental glaucoma model [79]. Moreover, the regulation of MSN phosphorylation is controlled by Rho-associated protein kinases (ROCK) [80], and specific ROCK inhibitors (e.g., Netarsudil), in turn, were currently launched as promising glaucoma medications [81]. All these findings emphasize the great potential of ANP32A as an auspicious drug target molecule in glaucoma therapy as well as for the treatment of other age-related neurodegenerative diseases. Nevertheless, the present study also indicates several limitations, which should be specifically addressed in future studies. In silico MD simulation provide great potential for the accurate prediction of bindings sites in PPIs but neglet the secondary structure instabilities of peptides in vivo. Modern technologies such as FRET or SPR might be used for the verification of this specific protein-peptide interaction in the future. However, targeted manipulation of the peptide binding motif (e.g., by alanine substitution) might be used for interaction validation with target protein ANP32A.

## 5. Conclusions

Overall, the present study comprehensively investigated the potential neuroprotective mechanisms of synthetic CDR1 (SCTGTSSDVGGYNYVSWYQ) on RGCs ex vivo and provided new insights into the complex molecular pathophysiology of glaucoma. Thereby, the synthetic CDR1 peptide specifically interacts with the acidic leucine-rich nuclear phosphoprotein 32A (ANP32A), which represents a highly abundant protein in the retina. In silico binding calculations predicted a significant binding site of synthetic CDR1 to the N-terminal LRR domain of ANP32A, which favors the formation of PPIs. In accordance, the expression levels of ANP32A-interacting proteins NME1 and PPP2R4 were also significantly down-regulated in the CDR1-treated retinae indicating impaired signaling transduction of ANP32A by synthetic CDR1. This might explain the CDR1-induced altered protein expression levels in the retinae involved in protein phosphorylation regulation (e.g., PPP2R4), acetyl-CoA biosynthesis (e.g., PDHA1) as well as cytoskeleton/membrane-associated proteins (e.g., MSN), which illustrate the potential neuroprotective modes of action of synthetic CDR1 ex vivo. Moreover, protein phosphorylation profiles as well as histone acetylation events, seem to be also affected by synthetic CDR1 in the retina ex vivo. However, further studies are needed to explore the biological function of ANP32A as a potential drug target in glaucoma therapy and to assess its medical application profile for clinical interventions in the future.

## Figures and Tables

**Figure 1 biomolecules-13-01161-f001:**
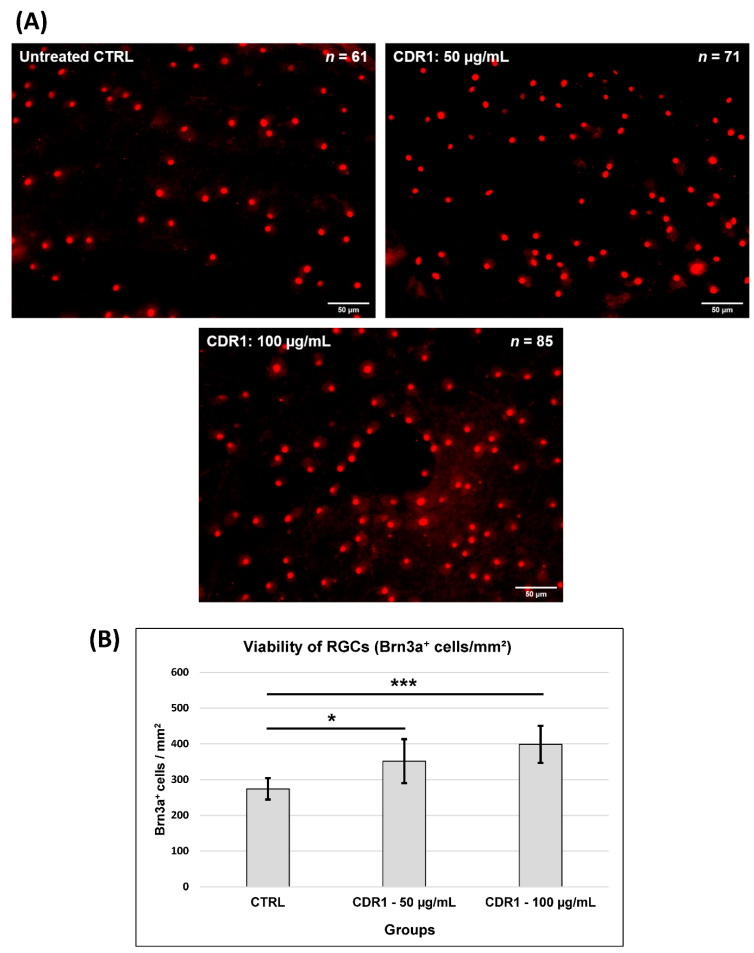
Neuroprotective effects of the synthetic CDR1 peptide (SCTGTSSDVGGYNYVSWYQ; 50 and 100 µL/mL) on the viability of RGCs ex vivo. (**A**) Exemplary high-resolution fluorescent images of retinal flat mounts ± CDR1 treatment (50 and 100 µL/mL) using Brn3a as a molecular marker for RGCs. The RGC density increased with ascending CDR1 peptide concentration (from 50 to 100 µg/mL). The number (*n*) of identified Brn3a^+^-RGCs is shown in each exemplary fluorescent image (**B**) Bar plot showing the number of Brn3a^+^ cells (RGCs/mm^2^) either in untreated retinal explants (CTRL) or treated with 50 and 100 µg/mL of synthetic CDR1. The RGC viability significantly increased in a concentration-dependent manner with the exogenous application of synthetic CDR1. (*n* = 6 for each group, *: *p* < 0.05 and ***: *p* < 0.001 in comparison to CTRL).

**Figure 2 biomolecules-13-01161-f002:**
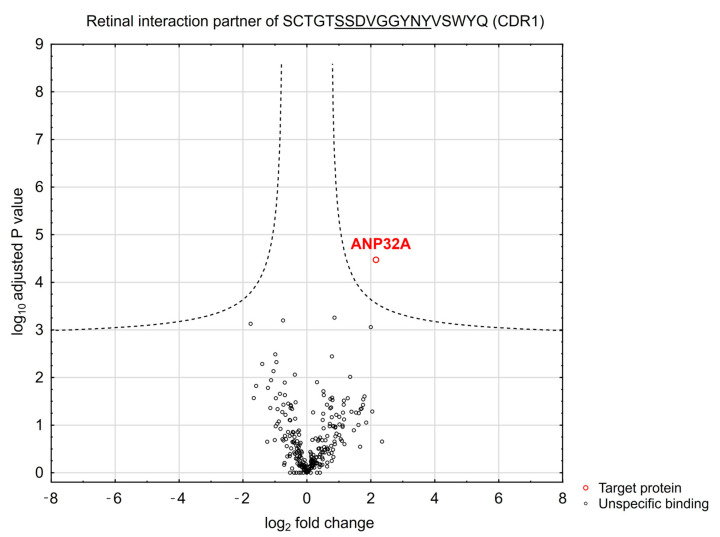
Identification of retinal protein interaction of synthetic CDR1 revealed by MS-based immunoprecipitation. The Volcano plot illustrates the log2 fold-change plotted against the −log10-adjusted *p* values of the proteins identified in both experimental groups (CDR1 and CTRL bead group). The acidic leucine-rich nuclear phosphoprotein 32A (ANP32A) was highly enriched in the CDR1-labeled bead group (SCTGTSSDVGGYNYVSWYQ) in comparison to unlabeled CTRL beads (*n* = 3 for each group; *p* < 0.001 and log2 fold change > 2).

**Figure 3 biomolecules-13-01161-f003:**
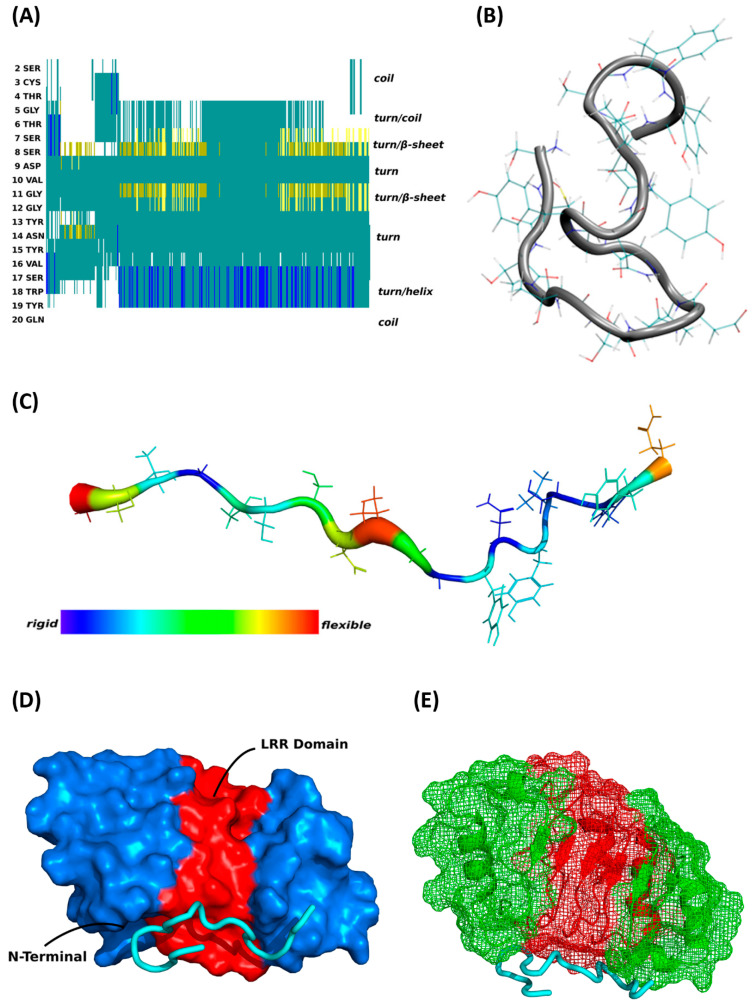
Secondary structure of the CDR1 peptide (SCTGTSSDVGGYNYVSWYQ) and the virtual docking analysis to retinal interaction partner acidic leucine-rich nuclear phosphoprotein 32A (ANP32A). (**A**) Structuring of the CDR1 peptide revealed by MD simulations. (**B**) Secondary structure of the CDR1 peptide exhibiting a β-hairpin like structure in the center position. The loop is partially formed by the CDR1 sequence motif S(6)-SDVG-G(11). (**C**) Flexibility of the CDR1 peptide along the peptide backbone fluctuations during MD simulations. The flexible portions of the sequence are dominated over three regions of the peptide, which are located in the center as well as one each at the terminal regions. (**D**,**E**) Potential binding site of the CDR1 peptide to retinal interaction partner ANP32A shown in cartoon and ribbon diagram. The peptide interacts with the leucine-rich repeat (LRR) of ANP32A, which is located at the N-terminal part of the protein (PDB protein structure: 4XOS). The N-terminal LRR domain of ANP32A is highlighted in red color in the protein structure from F(41) to K(111). The N- and C-Terminus of ANP32A are colored in blue in the cartoon diagram and in green in the ribbon 3D structure.

**Figure 4 biomolecules-13-01161-f004:**
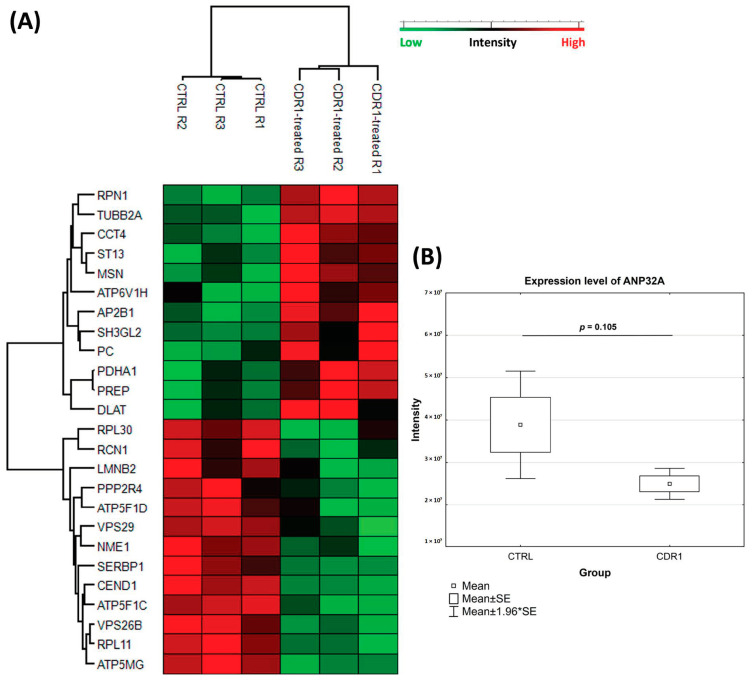
Quantitative proteomic analysis of CDR1-treated retinal explants (100 µg/mL) compared to untreated controls (CTRL). (**A**) Heatmap showing the expression levels of proteins, which were significantly differentially expressed between both experimental groups (CTRL and CDR1, *p* < 0.05). (**B**) Expression levels of acidic leucine-rich nuclear phosphoprotein 32A (ANP32A), which was identified as a potential interaction partner of synthetic CDR1. The protein levels of ANP32A were slightly diminished in the CDR1-treated retinal explants compared to untreated CTRL but were not supported by statistical significance (*p* = 0.105).

**Figure 5 biomolecules-13-01161-f005:**
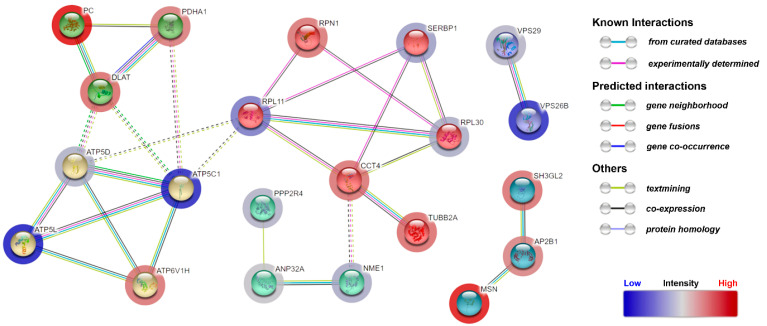
CDR1-induced signaling pathways in the retina. The protein interaction network illustrates the signaling cascades, which were elicited by CDR1 treatment in the retina during glaucomatous-like conditions ex vivo. STRING analysis revealed the signaling pathways and functions of the most affected proteins markers (*p* < 0.05) using a medium confidence score (0.4). The CDR1-specific target protein ANP32A shows a direct interaction with the proteins PPP2R4 and NME1. Proteins groups were assigned by MCL clustering, and protein intensities were labeled by color code (Red: ↑ and blue: ↓, CTRL vs. CDR1).

**Figure 6 biomolecules-13-01161-f006:**
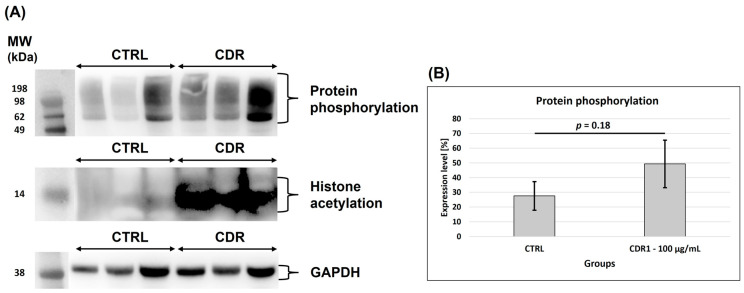
Protein phosphorylation and histone acetylation in retinal tissues ± CDR1 treatment. (**A**) Representative Western blots showing protein phosphorylation, histone acetylation, and the expression of housekeeping protein GAPDH in retinal tissues ± CDR1 treatment (100 µg/mL). Histone acetylation was clearly increased by CDR1 treatment compared to untreated CTRL but was quantitatively not analyzable. (**B**) Densitometric analysis of protein phosphorylation events in retinal tissues ± CDR1 (100 µg/mL). Band intensities were normalized to the housekeeping protein GAPDH. Error bars indicate standard deviation (SD). Protein phosphorylation is slightly increased by CDR1 treatment compared to untreated controls but was not supported by statistical significance (*p* = 0.18, *n* = 3 for each group).

**Table 1 biomolecules-13-01161-t001:** Protein markers were significantly up-regulated in CDR1-treated retinal explants compared to untreated controls (CTRL, *p* < 0.05).

No.	Protein ID	Protein Name	Gene Name	Score	Mol. Weight [kDa]	Peptides	Fold Change	*p*-Value	CDR1 vs. CTRL
1	P11498	Pyruvate carboxylase, mitochondrial	PC	46.55	129.6	3	2.99	<0.05	↑
2	P26038	Moesin	MSN	48.79	67.8	3	2.88	<0.05	↑
3	P50502	Hsc70-interacting protein	ST13	33.33	41.3	4	1.55	<0.05	↑
4	Q13885	Tubulin beta-2A chain	TUBB2A	65.75	49.9	21	1.52	<0.01	↑
5	P10515	Dihydrolipoyllysine-residue acetyltransferase component of PDH, mitochondrial	DLAT	9.34	69.0	3	1.51	<0.05	↑
6	Q99962	Endophilin-A1	SH3GL2	55.17	40.0	7	1.45	<0.05	↑
7	Q9UI12	V-type proton ATPase subunit H	ATP6V1H	128.60	55.9	8	1.43	<0.05	↑
8	P50991	T-complex protein 1 subunit delta	CCT4	50.63	57.9	7	1.41	<0.01	↑
9	Q9GMB0	Dolichyl-diphosphooligosaccharide--protein glycosyltransferase subunit 1	RPN1	23.80	68.7	8	1.39	<0.001	↑
10	P23687	Prolyl endopeptidase	PREP	137.82	80.8	11	1.37	<0.05	↑
11	P29804	Pyruvate dehydrogenase E1 component subunit α, mitochondrial	PDHA1	29.28	43.1	7	1.31	<0.05	↑
12	P63010	AP-2 complex subunit beta	AP2B1	65.96	104.6	11	1.16	<0.01	↑

↑: High abundant in the CDR1 group compared to CTRL. ↓: Low abundant in the CDR1 group compared to CTRL.

**Table 2 biomolecules-13-01161-t002:** Protein markers that were significantly down-regulated in CDR1-treated retinal explants compared untreated controls (CTRL, *p* < 0.05).

No.	Protein ID	Protein Name	Gene Name	Score	Mol. Weight [kDa]	Peptides	Fold Change	*p*-Value	CDR1 vs. CTRL
1	Q9UBQ0	Vacuolar protein sorting-associated protein 29	VPS29	38.05	20.5	2	1.33	<0.05	↓
2	Q15257	Serine/threonine-protein phosphatase 2A activator	PPP2R4	23.28	40.7	2	1.56	<0.05	↓
3	P30049	ATP synthase subunit delta, mitochondrial	ATP5F1D	60.95	17.5	2	1.58	<0.05	↓
4	Q03252	Lamin-B2	LMNB2	40.46	69.9	3	1.67	<0.05	↓
5	Q15293	Reticulocalbin-1	RCN1	66.83	38.9	3	1.68	<0.05	↓
6	P62888	60S ribosomal protein L30	RPL30	15.89	12.8	3	1.89	<0.05	↓
7	P15531	Nucleoside diphosphate kinase A	NME1	108.02	17.1	6	1.99	<0.05	↓
8	Q8NC51	Plasminogen activator inhibitor 1 RNA-binding protein	SERBP1	65.36	45.0	3	2.90	<0.05	↓
9	Q29205	60S ribosomal protein L11	RPL11	35.69	20.2	2	5.16	<0.01	↓
10	Q4G0F5	Vacuolar protein sorting-associated protein 26B	VPS26B	10.83	39.2	3	9.97	<0.01	↓
11	P36542	ATP synthase subunit gamma, mitochondrial	ATP5F1C	67.00	33.0	2	10.30	<0.01	↓
12	O75964	ATP synthase subunit g, mitochondrial	ATP5MG	37.14	11.4	2	11.08	<0.001	↓
13	Q29026	Cell cycle exit and neuronal differentiation protein 1	CEND1	66.37	14.0	2	11.81	<0.001	↓

↑: High abundant in CDR1 group compared to CTRL. ↓: Low abundant in CDR1 group compared to CTRL.

**Table 3 biomolecules-13-01161-t003:** List of CDR1-induced signaling pathways revealed by IPA analysis.

Signaling Pathways	−log (*p*-Value)	Molecules
Acetyl-CoA Biosynthesis I	4.62	DLAT, PDHA1
Mitochondrial Dysfunction	4.47	ATP5F1C, ATP5F1D, ATP5MG, PDHA1
Granzyme A Signaling	3.71	ANP32A, NME1
Oxidative Phosphorylation	3.62	ATP5F1C, ATP5F1D, ATP5MG
Mechanisms of Viral Exit from Host Cells	3.04	LMNB2, SH3GL2

## Data Availability

All MS data have been deposited to the ProteomeXchange Consortium via the PRIDE partner repository with the dataset identifier PXD038373.

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
