# Peer review of "Glaucoma-Associated CDR1 Peptide Promotes RGC Survival in Retinal Explants through Molecular Interaction with Acidic Leucine Rich Nuclear Phosphoprotein 32A (ANP32A)"

_biomolecules, 2023, doi:10.3390/biom13071161_

Round 1

Reviewer 1 Report

The present manuscript reports an interesting study carried out by the authors on a synthetic peptide (CDR1) able to promote the viability of retinal ganglion cells in ex vivo mammalian explants. The proteomic study identified the ANP32A protein as the molecular target through which the peptide exerts its neuroprotective activity. Aided by computational studies, the authors identified the interaction site of CDR1 in the LRR region of the ANP32A protein. The topic covered is certainly interesting and, according to the authors' idea, the CDR1 peptide could be a potential drug candidate for the therapeutic treatment of glaucoma.  At the present stage, the manuscript possesses elements of interest that could justify publication in this journal. However, there remain some aspects that authors should address:

1)    The authors must show the initial sequence they had chosen, better detail the problems encountered during the peptide synthesis, and how they came to decide the present sequence.

2)    It seems that the main experimental data supporting the interaction with the molecular target is the immunoprecipitation and the subsequent proteomic analysis. In this regard, similar control experiments using a scrambled sequence of CDR1 may help to validate what the authors claimed. Have the authors considered performing SPR and/or NMR experiments to quantify the degree of interaction between CDR1 and ANP32A? I think there is a possibility to obtain/purchase the recombinant protein.

3)    The conformational analysis of the CDR1 peptide can also be performed experimentally with the aid of circular dichroism spectroscopy.

4)    The discussion section appears extremely complex, with little connection with the experimental data and urges for simplification. Why, although the ANP32A protein has been indicated as the specific partner of CDR1, its expression is decreased following the treatment of neuronal cultures with the peptide? At what level does the peptide act? Instead, the discussions focus on the description of a vast network of interactions between proteins and consequent cascade effects that make this part meaningless and not so relevant for the robustness of the manuscript.

Author Response

The present manuscript reports an interesting study carried out by the authors on a synthetic peptide (CDR1) able to promote the viability of retinal ganglion cells in ex vivo mammalian explants. The proteomic study identified the ANP32A protein as the molecular target through which the peptide exerts its neuroprotective activity. Aided by computational studies, the authors identified the interaction site of CDR1 in the LRR region of the ANP32A protein. The topic covered is certainly interesting and, according to the authors' idea, the CDR1 peptide could be a potential drug candidate for the therapeutic treatment of glaucoma.  At the present stage, the manuscript possesses elements of interest that could justify publication in this journal. However, there remain some aspects that authors should address:

1)    The authors must show the initial sequence they had chosen, better detail the problems encountered during the peptide synthesis, and how they came to decide the present sequence.

Response: The initial peptide sequence was added in the manuscript. Unfortunately, we did not receive further details by the manufacturer about the synthesis problems. They just mentioned that the initial peptide sequence did not pass their quality criteria. Due to experiences from previous studies [1,2], we supposed that particularly the CDR1 sequence motif determines the biological function and activity of the peptide. That is the reason why we chose the present sequence.

2)    It seems that the main experimental data supporting the interaction with the molecular target is the immunoprecipitation and the subsequent proteomic analysis. In this regard, similar control experiments using a scrambled sequence of CDR1 may help to validate what the authors claimed. Have the authors considered performing SPR and/or NMR experiments to quantify the degree of interaction between CDR1 and ANP32A? I think there is a possibility to obtain/purchase the recombinant protein.

Response: We did not consider a scrambled CDR1 peptide for the immunoprecipitation experiment to avoid pseudo-affinities to other retinal protein targets. A targeted alanine substitution experiment might allow the identification of the specific CDR1 binding motif in future studies. We did not consider SPR/NMR experiments in this stage of the project, but might be an interesting topic for future collaboration projects. Unfortunately, our laboratory in not equipped for these kind of interaction experiments.

3)    The conformational analysis of the CDR1 peptide can also be performed experimentally with the aid of circular dichroism spectroscopy.

Response: We agree with the reviewer that CD spectroscopy could be also used for the conformational analysis of CDR1. However, the authors think that the MD simulations in combination with the docking analyses should provide suifficient structural information about synthetic CDR1 at this stage of the project. CD spectroscopy could be a promising technology to validate the conformation of synthetic CDR1 in future collaboration projects.

4)    The discussion section appears extremely complex, with little connection with the experimental data and urges for simplification. Why, although the ANP32A protein has been indicated as the specific partner of CDR1, its expression is decreased following the treatment of neuronal cultures with the peptide? At what level does the peptide act? Instead, the discussions focus on the description of a vast network of interactions between proteins and consequent cascade effects that make this part meaningless and not so relevant for the robustness of the manuscript.

Response: The CDR1 peptide seems to specifically interact with the LRR domain of ANP32A, which is an important binding for PPIs. Possibly, the blocking of this important binding site leads to enhanced proteasomal degradation of ANP32A as seen for its direct interaction partners NME1 and PPP2R4. Based on the experimental data, we suppose that the synthetic CDR1 peptide acts in the dose range from 50 to 100 µg/mL. The authors think that the discussion of the protein interaction networks is of high importance to understand the potential modes of action of synthetic CDR1. For sure, the exact modes of action have to be further validated in future studies, but the present data should be suifficient as first hints in the present project.

References

  1. Schmelter, C.; Fomo, K.N.; Perumal, N.; Manicam, C.; Bell, K.; Pfeiffer, N.; Grus, F.H. Synthetic Polyclonal-Derived CDR Peptides as an Innovative Strategy in Glaucoma Therapy. J. Clin. Med. 2019, 8, doi:10.3390/jcm8081222.
  2. Fomo, K.N.; Schmelter, C.; Atta, J.; Beutgen, V.M.; Schwarz, R.; Perumal, N.; Govind, G.; Speck, T.; Pfeiffer, N.; Grus, F.H. Synthetic antibody-derived immunopeptide provides neuroprotection in glaucoma through molecular interaction with retinal protein histone H3. 1. Frontiers in Medicine 2022, 3095.

Reviewer 2 Report

The authors explored the function of the synthetic CDR1 peptide on the viability of retinal ganglion cells in pig retinal explants in vitro. The authors further explored possible interaction partners through MS-based co-immunoprecipitation besides a proteomic analysis of treated and untreated retinal explants. The proteome indicates some signaling pathways related to CDR1, but no alteration in the expression of ANP32A. In addition, they explored phosphorylation and histone acetylation events upon CDR1 treatment. The novelty is the demonstration that the synthetic CDR1 peptide has a neuroprotective effect on RGC degeneration in retinal explants and the identification of ANP32A as a strong ligand. The proteome analysis was interesting but did not corroborate much for the pathway further analyzed. And western blot was poorly represented.

Overall, the results are interesting, but the manuscript needs some improvements in order to be published. Some suggestions are described below:

1.     Lines 53-54: Please add reference.

2.     Lines 70, 76 and others: in vivo in vitro and ex vivo in italic.

3.     Line 88: reference 18 does not match the sentence information.

4.     Lines 112-114: 1st sentence should be in the introduction section.

5.     Line 127: The retinal explants ex vivo is not a model of glaucoma, but of RGC degeneration.

6.     Please add the catalog number of the commercial products in the Methods sections.

7.     Line 144 “Both concentrations were revealed from a previous experiment (data not shown) and selected as starting point for the present study” – please add the data as a supplementary figure in order to justify the selected concentrations.

8.     Line 146: what does it mean n=6? 6 pigs or 6 explants? Please describe it better.

9.     Figure 4: Low-High scale, the numbers cannot be seen, Please, adjust or remove it.

10.  Figure 6: Please change the histone acetylation blot. The blot presented seems to be poorly processed. 

11.  It is not discussed how the synthetic CDR1 peptide would enter the cell to ligate with ANP32A to promote its activities through this ligation. Please comment.

12.  Line 673, the conclusion sentence needs to be adjusted as a “possibility of action of the synthetic CDR1”- the result presented are not conclusive but suggestive.

13.  There is a lot of self-citation.

Author Response

The authors explored the function of the synthetic CDR1 peptide on the viability of retinal ganglion cells in pig retinal explants in vitro. The authors further explored possible interaction partners through MS-based co-immunoprecipitation besides a proteomic analysis of treated and untreated retinal explants. The proteome indicates some signaling pathways related to CDR1, but no alteration in the expression of ANP32A. In addition, they explored phosphorylation and histone acetylation events upon CDR1 treatment. The novelty is the demonstration that the synthetic CDR1 peptide has a neuroprotective effect on RGC degeneration in retinal explants and the identification of ANP32A as a strong ligand. The proteome analysis was interesting but did not corroborate much for the pathway further analyzed. And western blot was poorly represented.

Overall, the results are interesting, but the manuscript needs some improvements in order to be published. Some suggestions are described below:

  1. Lines 53-54: Please add reference.

Response: The respective reference was added in the manuscript.

  1. Lines 70, 76 and others: in vivo in vitro and ex vivo in italic.

Response: The publisher does not support the italic style for these expressions.

  1. Line 88: reference 18 does not match the sentence information.

Response: Hepatopoietin is another name for the protein ANP32A and describes the same molecule.

  1. Lines 112-114: 1st sentence should be in the introduction section.

Response: The sentence was added in the introduction section in line 100-102.

  1. Line 127: The retinal explants ex vivo is not a model of glaucoma, but of RGC degeneration.

Response: The sentence was revised according to the recommendation of the rewiever.

  1. Please add the catalog number of the commercial products in the Methods sections.

Response: We added the catalog numbers for the commercial products in the manuscript.

  1. Line 144 “Both concentrations were revealed from a previous experiment (data not shown) and selected as starting point for the present study” – please add the data as a supplementary figure in order to justify the selected concentrations.

Response: We added the additional data in the respective section of the manuscript.

  1. Line 146: what does it mean n=6? 6 pigs or 6 explants? Please describe it better.

Response: Missing information was added in the method section.

  1. Figure 4: Low-High scale, the numbers cannot be seen, Please, adjust or remove it.

Response: Low-high scaling was removed from Fig. 4

  1. Figure 6: Please change the histone acetylation blot. The blot presented seems to be poorly processed.

Response: We processed the blot of the histone acetylation as good as possible and provided the unprocessed image in the supplementary. Moreover, we mentioned in the method section (line 593-598) that this observation is just a first hint for increased histone acetylation events by synthetic CDR1 and should be confirmed in future studies 

  1. It is not discussed how the synthetic CDR1 peptide would enter the cell to ligate with ANP32A to promote its activities through this ligation. Please comment.

Response: In literature it has been reported that peptides are able to penetrate the cell membrane by direct translocation as well as by receptor-mediated endocytosis (e.g., clathrin-mediated endocytosis) [1], which might be also assumed as potential mechanism for the synthetic CDR1 peptide. This topic was shortly discussed in the manuscript in line 539-542.

  1. Line 673, the conclusion sentence needs to be adjusted as a “possibility of action of the synthetic CDR1”- the result presented are not conclusive but suggestive.

Response: We adjusted the conclusion section according to the recommendation of the reviewer.

  1. There is a lot of self-citation.

Response: We tried to reduce this issue within the manuscript.

References

  1. Xie, J.; Bi, Y.; Zhang, H.; Dong, S.; Teng, L.; Lee, R.J.; Yang, Z. Cell-Penetrating Peptides in Diagnosis and Treatment of Human Diseases: From Preclinical Research to Clinical Application. Front. Pharmacol. 2020, 11, 697, doi:10.3389/fphar.2020.00697.

Reviewer 3 Report

The paper “Glaucoma-associated CDR1 peptide promotes RGC survival in retinal explants through molecular interaction with acidic 3 leucine rich nuclear phosphoprotein 32A (ANP32A)” is an interesting and valuable paper that shows a new innovative approach to the treatment of glaucoma. To date, there is no adequate treatment option that directly affects the neurodegenerative molecular mechanism of glaucoma and specific drug target molecules for therapy are still lacking. Neuroprotection as the goal of treatment represents a step forward in the treatment of this disease, since only in this way can we preserve vision in the long term.

A great effort has been put into this research, the methods are described in detail, the results are clearly presented and the conclusion is logically derived. This study comprehensively investigated the neuroprotective mechanisms of synthetic CDR1 and provides new insights into the complex molecular pathophysiology of glaucoma. Synthetic CDR1 peptide provides great potential for glaucoma treatment in the future by inducing its neuroprotective mechanism through specific interaction with the N terminal LRR domain of ANP32A.

Author Response

No revisions were needed. We thank the reviewer for the positive feedback.

Round 2

Reviewer 1 Report

Based on the answers provided by the Authors I have to conclude that they consider the manuscript self-consistent in the present form. I can understand the motivations put forward by the Authors but, as things stand, there's no reason I change my original judgment on the quality of the manuscript. I therefore leave the decision to proceed with the publication of this manuscript to the Editor.

Regards

Author Response

No revisions were needed. We thank the reviewer for the feedback.

Reviewer 2 Report

The authors responded to almost all requested pointsbut it still needs some improvements to be published. Some suggestions are described below:

1.     There is an “in vivo“ with the italic format in line 72. Please, apply this configuration to all other in vivoex vivoand in vitro expressions.

2.     The sentence in line 127 has been corrected. However, in line 350 the research model has still been cited as a “glaucoma model”. Please, revise and change it.

3.     Figure S1 has been added, but it is still unclear why the 50 µg and 100 µg concentrations were chosen. Please, describe it better.

4.     Line 146, if each of the 6 retinal explants belongs to a different pig eye, then n=6, please correct it.

5.     Figure 6, the histone acetylation blott needs to be improved. The western blotting profile of the antibody on its datasheet or in other papers does not contain the unspecific labeling observed in this figure. Another western-blot must be presented.

6.     The sentence is missing parenthesis in lines 571-572. Please, revise.

7.     There are still a lot of self-citations.

Author Response

The authors responded to almost all requested points, but it still needs some improvements to be published. Some suggestions are described below:

  1. There is an “in vivo“ with the italic format in line 72. Please, apply this configuration to all other in vivoex vivoand in vitroexpressions.

Response: The manuscript was revised according to the recommendation of the reviewer.

  1. The sentence in line 127 has been corrected. However, in line 350 the research model has still been cited as a “glaucoma model”. Please, revise and change it.

Response: The manuscript was revised according to the recommendation of the reviewer.

  1. Figure S1 has been added, but it is still unclear why the 50 µg and 100 µg concentrations were chosen. Please, describe it better.

Response: The explanation for choosing these two concentrations was added in line 151-154 of the revised manuscript.

  1. Line 146, if each of the 6 retinal explants belongs to a different pig eye, then n=6, please correct it.

Response: The retinal explants were taken from different pig eyes. The manuscript was revised accordingly.

  1. Figure 6, the histone acetylation blott needs to be improved. The western blotting profile of the antibody on its datasheet or in other papers does not contain the unspecific labeling observed in this figure. Another western-blot must be presented.

Response: We tried to revise the figure of the Western blot by adjusting the auto-contrast option. Unfortunately, it is not possible to repeat the Western Blot analysis due to sample material limitations. However, the authors think that there is a clear chemiluminescent signal in the CDR1-treated replicates in the expected mass range (around 14 kDa). Moreover, we also clarified in the manuscript that this finding is just a first hint, but should be further investigated/verified in future studies.

  1. The sentence is missing parenthesis in lines 571-572. Please, revise.

Response: The manuscript was revised according to the recommendation of the reviewer.

  1. There are still a lot of self-citations

Response: We tried to address this issue in the revised version of the manuscript.